# EVF-SAM: Early Vision-Language Fusion for Text-Prompted Segment Anything Model

## Abstract

Segment Anything Model (SAM) has attracted widespread attention for its superior interactive segmentation capabilities with visual prompts while lacking further exploration of text prompts. In this paper, we empirically investigate what text prompt encoders (*e.g.*, CLIP or LLM) are good for adapting SAM for referring expression segmentation and introduce the *Early Vision-language Fusion-based* SAM (**EVF-SAM**). EVF-SAM is a simple yet effective referring segmentation method which exploits multimodal prompts (*i.e.*, image and text) and comprises a pre-trained vision-language model to generate referring prompts and a SAM for segmentation. Surprisingly, we observe that: (1) multimodal prompts and (2) vision-language models with early fusion (*e.g.*, BEIT-3) are beneficial for prompting SAM for accurate referring segmentation. Our experiments show that the proposed EVF-SAM based on BEIT-3 can obtain state-of-the-art performance on RefCOCO/+/g for referring expression segmentation and demonstrate the superiority of prompting SAM with early vision-language fusion. In addition, the proposed EVF-SAM with 1.32B parameters achieves remarkably higher performance while reducing nearly 82% of parameters compared to previous SAM methods based on large multimodal models. Code and models will be made publicly available.

## 1 Introduction

Segment Anything Model (SAM) (Kirillov et al., 2023) brings interactive segmentation paradigm to public view. Well-trained on the SA-1B dataset, SAM achieves stunning performance and quickly becomes popular as a vision foundation model for object localization and beyond. Various SAM variants (Xiong et al., 2023; Zhang et al., 2023; Zhao et al., 2023; Ke et al., 2024) have been explored, achieving better efficiency or higher precision. Despite SAM's surprising abilities like point-prompted and box-prompted segmentation, it is a pity that the text-prompted segmentation ability remains conceptual. We retrospect such task to Referring Expression Segmentation (RES). RES focuses on the solution that one predicts the segmentation mask according to the text description given by users, which enjoys several explorations by some traditional end-

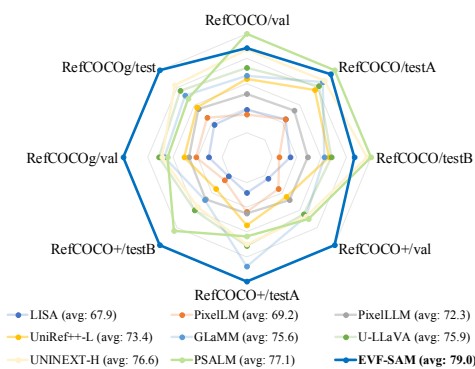

Figure 1: EVF-SAM achieves competitive performance among various benchmarks for referring expression segmentation.

to-end models (Hu et al., 2016; Liu et al., 2017; Shi et al., 2018; Chen et al., 2019; Ye et al., 2019; Hu et al., 2020; Ding et al., 2021; Li & Sigal, 2021b; Wang et al., 2022b; Yang et al., 2022; Liu et al., 2023c; Wu et al., 2023; Liu et al., 2023e; Yan et al., 2023), and is broadened by some Large Multimodal Models (LMM) (Lai et al., 2023; Yang et al., 2023; Ren et al., 2023; Pi et al., 2023; Xu et al., 2023; Zhang et al., 2024; Xia et al., 2023; Rasheed et al., 2023).

The key challenge lies in empowering SAM with language understanding ability for segmentation according to text prompts, *e.g.*, referring expression segmentation. Fig. 2 summarizes previous works which explore the text-prompted abilities of SAM: (a) SAM with grounded detector: A two-stage framework where a grounded detector generates a bounding box to prompt SAM, *e.g.*,

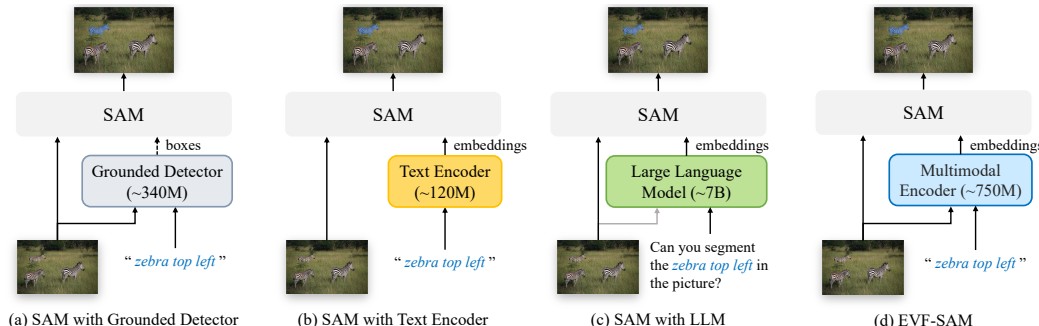

(a) SAM with Grounded Detector  (b) SAM with Text Encoder  (c) SAM with LLM  (d) EVF-SAM

Figure 2: **Comparisons of different Text-prompted SAM.** (a) Given input texts, several works (Ren et al., 2024) leverage grounded detectors, *e.g.*, Grounding DINO (Liu et al., 2023d), to generate box prompts for SAM. (b) A natural idea to support text prompts is to use an *off-the-shelf* text encoder to generate text embeddings for SAM (Kirillov et al., 2023; Li et al., 2023) while the performance of referring segmentation is inferior. (c) Several works (Lai et al., 2023; Yang et al., 2023; Rasheed et al., 2023) adopt Large Language Models (LLM) or Large Multimodal Models (LMM) to generate prompt embeddings for SAM in an autoregressive manner, which incurs a large computation burden. (d) Our proposed EVF-SAM exploits an effective Multimodal Encoder for text-prompted SAM with higher performance and fewer parameters compared to LLM-based methods.

Grounded-SAM (Ren et al., 2024). However, those methods suffer from a sub-optimal architecture, where segmentation heavily relies on the accuracy of the detector, and it is difficult to optimize due to its non-end-to-end nature. (b) SAM with text encoder: A *off-the-shelf* text encoder, *e.g.*, CLIP (Radford et al., 2021), is used to encode the text prompt, providing text embeddings for SAM. Whereas the semantic gap exists between the text embeddings and SAM which is pre-trained with geometric prompts, *i.e.*, points or boxes, thus the segmentation performance is inferior. (c) SAM with LLM: A Large Language Model (LLM) (or Large Multimodal Model) is employed and fine-tuned to get the desired embeddings about object information. The embeddings will be used to predict segmentation masks based on image features. However, these LLM-based models are often computationally expensive, requiring massive memory and computation budgets, and the training is challenging. Additionally, complex conversation templates need to be manually designed to instruct the LLM for referring segmentation. Can we leverage a more efficient but effective method to empower SAM with text-prompted ability in an end-to-end manner?

To this end, we empirically investigate how to encode text prompts for SAM to address referring expression segmentation. Interestingly, we observe that (1) using multimodal prompts including both the text and image performs better than the text-only prompts and (2) the Multimodal Encoders with early vision-language fusion demonstrate significant superiority compared to text-only encoders or Large Language Models, as shown in Fig. 2 (d).

Motivated by the above observations, we extend SAM for language understanding and text-prompt capabilities by incorporating a Multimodal Encoder with **E**arly **V**ision-Language **F**usion (EVF) and present EVF-SAM in this paper. The proposed EVF-SAM aims to be a simple framework to prompt SAM with texts and illustrate how to prompt SAM to follow referring expressions effectively. EVF-SAM is built on the *off-the-shelf* foundation models and comprises a Multimodal Encoder, an early-fused vision-language model, *e.g.*, BEIT-3 (Wang et al., 2022a), and a simple projector to generate prompt embeddings for SAM. EVF-SAM does not include elaborate designs or modules and is easy for scaling to larger models.

Training EVF-SAM is simple and conducted on referring segmentation datasets, *e.g.*, RefCOCO (Yu et al., 2016), which is appropriate to adapt the original SAM for text prompts. Despite the simple architecture, our EVF-SAM achieves superior performance on referring expression segmentation tasks and outperforms previous attempts with Large Language Models (Lai et al., 2023; Yang et al., 2023; Rasheed et al., 2023), as shown in Fig. 1. The experimental results demonstrate that (1) using a multimodal encoder with the input text and image and (2) early fusion between the text and image contribute to the better-referring ability for SAM, showing a promising direction for text-prompted SAM. Additionally, the experiments also show the superiority of our EVF-SAM using a multimodal encoder over previous methods with decoder-only Large Language Models: (1) EVF-SAM reduces

huge amounts of parameters, *e.g.*, 82% parameters compared to LISA; (2) EVF-SAM relies less on handcrafted templates or instructions, which is more efficient and flexible; (3) EVF-SAM obtains better performance with less training data.

Our main contributions can be summarized as follows:

- We investigate the most effective approach to prompt SAM with texts by leveraging the Multimodal Encoder with multimodal inputs and the early vision-language fusion, which outperforms vanilla text encoders or Large Language Models.

- We formulate the paradigm for text-prompted SAM and propose EVF-SAM, which is modular and readily integrated with mainstream foundation models. In addition, EVF-SAM gets rid of hand-crafted templates, and the training is stable and efficient compared to methods using Large Language Models.

- The proposed EVF-SAM, only trained with open-source datasets, achieves state-of-the-art performance on the referring expression segmentation tasks, *i.e.*, RefCOCO/+/g, demonstrating the effectiveness of our paradigm. Notably, EVF-SAM reduces parameters by 82% (1.3B *v.s.* 7.7B) compared to previous works based on Large Language Models.

## 2 RELATED WORK

### 2.1 TEXT-PROMPTED SEGMENT ANYTHING MODELS

**Segment Anything Model.** SAM (Kirillov et al., 2023) is an interactive segmentation model capable of predicting non-semantic masks based on various types of prompts (points, boxes, coarse masks). Trained on a large-scale dataset, SAM demonstrates strong generalization capability for segmenting diverse common objects. Several works (Xiong et al., 2023; Zhao et al., 2023) address the massive computation cost of SAM and propose efficient variants. Efficient-SAM (Xiong et al., 2023) distils the image encoder of SAM, achieving comparable performance with significantly fewer parameters. Fast-SAM (Zhao et al., 2023), leveraging the YOLOv8 (Jocher et al., 2023) architecture, achieves a $50\times$ speedup for inference. SAM-HQ (Ke et al., 2024) addresses the segmentation quality of SAM and utilizes low-level features from the image encoder to enhance the mask decoder for better accuracy. Although SAM excels in visual-based segmentation tasks with box/point/mask prompts, it currently lacks language understanding abilities and it's infeasible to directly use text prompts for referring segmentation or semantic segmentation.

**Text-Prompted explorations.** Recently, several works (Ren et al., 2024; Zhao et al., 2023; Li et al., 2023) have explored text prompts for SAM to segment objects according to the instructions or referring expressions. Grounded-SAM (Ren et al., 2024) leverages the Grounding DINO (Liu et al., 2023d) to obtain text-prompted boxes and feed the boxes to SAM for segmentation results, which formulates the non-end-to-end two-stage frameworks. Fast-SAM (Zhao et al., 2023) matches the similarity of CLIP (Radford et al., 2021) features between the text and Region of Interest (RoI) of image. RefSAM (Li et al., 2023) employs a lightweight cross-modal MLP to project the text embeddings of the referring expressions into SAM's sparse embeddings and dense embeddings. LISA (Lai et al., 2023; Yang et al., 2023) employs a Large Multimodal Model, *e.g.*, LLaVA (Liu et al., 2023b) to extract multimodal embeddings for SAM through the auto-regressive decoder. The aforementioned methods either suffer from poor performance or are computationally expensive. Referring expression segmentation based on SAM is a promising area for exploration, offering significant potential. We propose an effective end-to-end model that overcomes SAM's limitations by enabling text-prompted segmentation capabilities.

### 2.2 REFERRING EXPRESSION SEGMENTATION

Referring Expression Segmentation (RES) is a multimodal segmentation task requiring accurate pixel-wise segmentation and fine-grained language understanding.

**Referring Segmentation via Text Encoders.** Prevalent methods (Li & Sigal, 2021b; Wang et al., 2022b; Yang et al., 2022; Liu et al., 2023c) tend to leverage transformer-based text encoders, *e.g.*, BERT (Devlin et al., 2018) or CLIP (Radford et al., 2021), to encode expression texts into embeddings as guidance for segmentation. RefTr (Li & Sigal, 2021a) uses a visual-language encoder

to fuse image and text features and regresses the box and mask with a carefully designed query processor. LAVT (Yang et al., 2022) leverages a hierarchical Vision Transformer (Dosovitskiy et al., 2020) (ViT) to perform language-aware visual encoding. CRIS (Wang et al., 2022b) designs a vision-language decoder to merge CLIP features, propagating fine-grained semantic information from textual representations to each pixel-level activation. PolyFormer (Liu et al., 2023c) follows the encoder-decoder structure, employing a transformer decoder to generate regression results. Novel methods pay attention to being compatible with multiple tasks to formulate a uniform model. UNINEXT (Yan et al., 2023), UniRef++ (Wu et al., 2023) and UniLSeg (Liu et al., 2023e) employ similar frameworks but focus on utilizing datasets from different fields to empower their generalization capability. Although these traditional models are usually lightweight and achieve fine performance, They fail to integrate with large-scale foundation models, *e.g.*, SAM(Kirillov et al., 2023), LLaVA(Liu et al., 2023b), thereby struggling to keep pace with the trend of increasingly extensive pre-training.

**Referring Segmentation via Large Language Models.** In the context of the rapid development of Large Multimodal Models (Liu et al., 2023b; Bai et al., 2023; Sun et al., 2023b;a) (LMM), a number of works (Lai et al., 2023; Yang et al., 2023; Ren et al., 2023; Pi et al., 2023; Xu et al., 2023; Zhang et al., 2024) have leveraged these models to encode expression texts for referring expression segmentation tasks. LISA (Lai et al., 2023; Yang et al., 2023) finetune LLaVA (Liu et al., 2023b) to make it able to answer questions related to segmentation with a fixed template like 'It is [SEG].', where the hidden embeddings at the place of special token [SEG] will be seen as multimodal features extracted by LMM. PixelLM (Ren et al., 2023) extends LISA by building a segmentation codebook to enable multi-object segmentation. PixelLLM (Xu et al., 2024) empowers the vision-language model to take locations (*e.g.*, a set of points or boxes) as either inputs or outputs. PerceptionGPT (Pi et al., 2023) proposes an end-to-end architecture. u-LLaVA (Xu et al., 2023) supports multi-task. PSALM (Zhang et al., 2024) imports mask tokens to LMM input for better performance. However, those methods tend to adopt heavy architectures, especially the LLMs or LMMs, leading to a heavy computation burden for downstream applications. In contrast, we find that lightweight vision-language models perform better for encode text prompts for referring image segmentation.

## 3 METHOD

### 3.1 MOTIVATION: SAM WITH VISION-LANGUAGE MODELS

Considering that SAM (Kirillov et al., 2023) has a strong generalization capability for image segmentation while the text-prompted ability has not been revealed, we investigate how to encode text prompts for SAM in this section. We started by using the vanilla text encoder, as shown in Fig. 3 and conducted preliminary experiments on RefCOCO (testA) to evaluate the referring ability of SAM, shown in Tab. 1.

**Multimodal referring information for SAM.** SAM (Kirillov et al., 2023) has explored the feasibility of employing a CLIP text encoder to facilitate text-prompted segmentation, as illustrated in Fig. 3 (a). We owe its weak perfor-

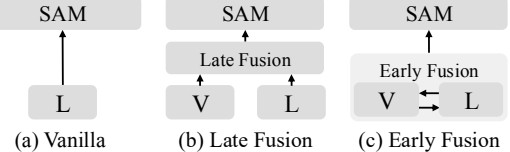

(a) Vanilla     (b) Late Fusion     (c) Early Fusion

Figure 3: **Architectural explorations for text-prompted SAM.** 'L' and 'V' denote the text encoder and vision encoder. We mainly explore three schemes: (a) vanilla baseline with a simple text encoder, (b) multimodal inputs with a late fusion, *i.e.*, concatenation, and (c) multimodal inputs with early vision-language fusions, *e.g.*, BEIT-3 (Wang et al., 2022a).

mance to the single-modal referring information. CLIP-prompted SAM achieves 63.4 cIoU at the RefCOCO/testA benchmark, far from well-defined baselines. CLIP exhibits strong alignment between text and image modalities, this alignment is insufficient for fine-grained tasks like segmentation. The referring information extractor should be provided with the input image and text prompts to ensure accurate alignment between the text expression and the relevant image region. We observe performance improvements after using multimodal prompts, *i.e.*, 63.4 *v.s.* 67.9 for CLIP and 65.1 *v.s.* 83.7 for BEIT-3.

**Early-fused architecture.** Some existing works, *e.g.*, LISA (Lai et al., 2023; Yang et al., 2023), UniLSeg (Liu et al., 2023e), advocate for fusing visual and textual information simply before the

Table 1: **Motivation analysis.** Both CLIP and BEIT-3 are of `Large` scale, with comparable numbers of parameters. Specifically, CLIP has a total parameter count of 428M, while BEIT-3 totals 673M parameters. Metric of LLaVA (Liu et al., 2023b) is borrowed from LISA-7B (Lai et al., 2023)

|  | CLIP (Text) | CLIP (Text+Image) | BEIT-3 (Text) | BEIT-3 (Text+Image) | LLaVA (Text+Image) |
|---|---|---|---|---|---|
| cIoU (RefCOCO) | 63.4 | 67.9 | 65.1 | **83.7** | 79.1 |

mask generator and are widely considered as 'early fusion'. However, we argue that these approaches are not early enough. As illustrated in Fig. 3 (b), we define such fusion for separately encoded single-modal prompts as 'late fusion'. In contrast, as shown in Fig. 3 (c), we define the fusion during feature extraction, where both modalities can access the dense information of the other one, as 'early fusion', *e.g.*, ViLT (Kim et al., 2021), BEIT-3 (Wang et al., 2022a), which incorporate the cross-modal fusions within the encoder.

We leverage the 'early-fusion' vision-language model as the Multimodal Encoder to generate prompt embeddings for SAM. Tab. 1 shows that our investigation indicates that early-fusion outperforms late-fusion, *i.e.*, 83.7 for BEIT-3 and 67.9 for CLIP. We believe the early-fused architecture, as defined by our approach, is beneficial for encoding text prompts since the cross-modal fusions will further enhance the semantic representation for text embeddings. In addition, the text-to-image fusions guide the image branch to aggregate features which are aligned with text prompts, making the output embeddings more accurate for prompt SAM.

**Encoder-based feature extractor.** Recently, LISA (Lai et al., 2023; Yang et al., 2023) and several LLM-based methods (Rasheed et al., 2023; Ren et al., 2023) acquire the prompt embeddings for SAM with a special token through the auto-regressive generation. However, the uncontrolled length of the answering query introduces instability during both training and inference. Forcing the model to conform to a specific answering template can lead to language drift. In contrast, encoder-based architectures can maintain a consistent sequence length of inputs and outputs. Utilizing the encoder-based method not only offers convenience but also yields superior performance, *i.e.*, 79.1 for LLaVA and 83.7 for BEIT-3. Notably, the encoder-based text-prompted SAM will reduce a massive computation burden compared to the LLM-based methods.

## 3.2 ARCHITECTURE

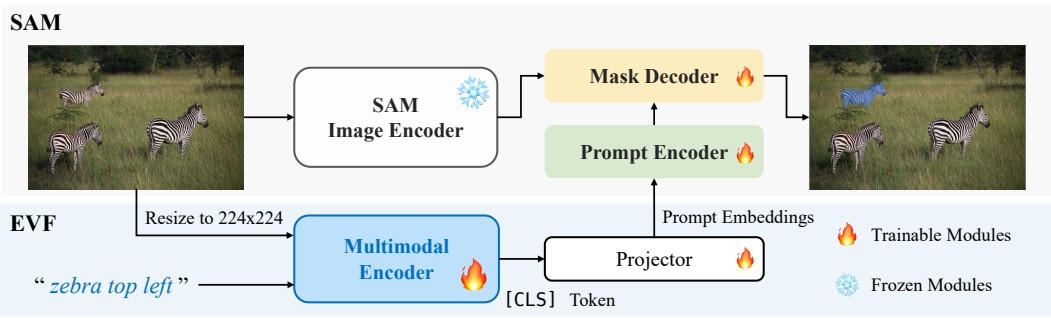

Figure 4: **The overall architecture of EVF-SAM.** The proposed EVF-SAM maintains the original architecture of SAM and keeps the weights of the SAM Image Encoder frozen. EVF-SAM exploits the Multimodal Encoder with Early Vision-Language Fusion (EVF) to encode both text prompts and the low-resolution input image (which is resized to $224 \times 224$). Then the output `[CLS]` token is projected as prompt embeddings and fed into the prompt encoder of SAM for generating the referring segmentation results.

Fig. 4 illustrates the overview of EVF-SAM, which is a simple yet effective framework with three modules: Multimodal Encoder, Projector, and Segment Anything Model (SAM).

**Multimodal Encoder.** The Early Vision-Language Fused encoder adopts the input image and text and outputs fused multimodal embeddings. In EVF-SAM, we mainly adopt BEIT-3 (Wang et al., 2022a) as the Multimodal Encoder, which formulates a multi-way transformer. The text is tokenized by XLMRobertaTokenizer (Conneau et al., 2019) while the image is resized to $224^2$ and patched by

a 1/16 convolution layer. Within each block of the encoder, the image and text tokens will be fused in the attention block and then fed into separate Feed-Forward Networks (FFN). We follow ViT (Dosovitskiy et al., 2020) to retrieve the `[CLS]` token as the output multimodal embeddings.

**Projector.** Different foundation models tend to have different embedding dimensions (1024 for BEIT-3-Large, 768 for BEIT-3-Base, and 256 for SAM mask decoder). We adopt a simple MLP projector containing 2 Linear layers, activated by ReLU. In EVF-SAM, we do not design elaborate modules for better performance due to the following reasons: (1) the simple MLP is effective enough (Liu et al., 2023b; Kim et al., 2021), (2) using MLP is efficient for training and inference, and (3) the simple projector will have few impacts on the pre-trained knowledge of foundation models.

**Adapted prompt encoder for SAM.** SAM contains 3 main modules: (a) Image Encoder: a Vision Transformer (Li et al., 2022) (ViT), extracting fine-grained feature maps from the input image. (b) Prompt Encoder: receiving interactive prompts and encoding them into hidden embeddings. (c) Mask Decoder: a lightweight mask generator to output the final masks based on previous embeddings. In EVF-SAM, we maintain the architecture of the image encoder and mask decoder while extending the prompt encoder to further gather the embeddings from the Multimodal Encoder. Specifically, the original prompt encoder encodes point or box prompts to *sparse embeddings* of $R^{B \times N \times D}$, where $B$, $N$, and $D$ refer to the batch size, number of points/boxes, and the embedding dimension, respectively. In EVF-SAM, the projected multimodal embeddings of $R^{B \times 1 \times D}$ from the Multimodal Encoder will be concatenated to a zero-initialized *sparse embeddings* and then fed into the mask decoder.

### 3.3 Training

**Instruction template-free.** In most LLM-based frameworks, *e.g.*, LISA (Lai et al., 2023; Yang et al., 2023), instruction templates are required to prompt the Large Multimodal Models (LMM) for the segmentation task, *e.g.*, '*Can you segment {object} in the picture*' with answer '*It is [SEG].*'. Removing instruction templates will affect the performance of LMMs, therefore, users need to follow the corresponding templates for referring image segmentation. In contrast, EVF-SAM does not require pre-training on QA (question-answering) datasets, thus eliminating the need for instruction templates. We adopt the expression phrases or sentences as input. This template-free approach simplifies training and inference.

**Trainable modules.** The Multimodal Encoder (EVF) is fully trainable during our training process, allowing it to learn how to generate multimodal embeddings tailored for SAM, which requires sufficient localization information for segmentation. For SAM, we keep the image encoder frozen during training while we enable training for the prompt encoder and mask decoder. Our experiments revealed that freezing the prompt encoder and mask decoder only leads to a minimal performance drop while maintaining SAM's ability. We present details in Sec. 4.4.

**Unified training with multi-tasks.** To further enhance the generic multi-task segmentation capabilities of EVF-SAM, including semantic segmentation and fine-grained part segmentation, we present a unified training strategy for EVF-SAM with diverse training datasets, such as ADE20K (Zhou et al., 2017b), PartImageNet (He et al., 2022) and PASCAL-Part (Chen et al., 2014).

However, we observe a performance degradation when simply mixing the training data of referring and semantic segmentation (shown in Tab. 7 of the Appendix), which can be attributed to the semantic conflict among different tasks, as discussed in UniLSeg (Liu et al., 2023e). To alleviate the aforementioned conflicts, we leverage a special text token `[semantic]` and input '`[semantic]`{*category*}' for semantic/part segmentation.

## 4 Experiments

### 4.1 Datasets and Metrics

**Datasets.** We mainly conduct the experiments on RefCLEF (Kazemzadeh et al., 2014), RefCOCO, RefCOCO+ (Yu et al., 2016; Kazemzadeh et al., 2014), and RefCOCOg (Nagaraja et al., 2016; Mao et al., 2016). Specifically, RefCOCOg contains longer expressions which are manually anno-

Table 2: Comparison of cIoU on different benchmarks between our proposed EVF-SAM and previous state-of-the-art methods. **Bold**: the best results. Underline: the second-best results. AVG represents the average metric across the eight RefCOCO-series benchmarks. We abbreviate the datasets: COCO (C) (Lin et al., 2014), RefCOCO (RC) (Yu et al., 2016; Nagaraja et al., 2016; Mao et al., 2016; Kazemzadeh et al., 2014), Objects365 (O) (Shao et al., 2019), Video segmentation datasets (V), ADE20K (A) (Zhou et al., 2017a; 2019), COCO-Stuff (CS) (Caesar et al., 2018), PACO-LVIS (PL) (Ramanathan et al., 2023), PASCAL-Part (PP) (Chen et al., 2014), GranD (G) (Rasheed et al., 2023), PASCAL VOC2010 (PV) (Everingham et al., 2010), MUSE (M) (Ren et al., 2023), gRefCOCO (gRC) (Liu et al., 2023a), COCO-Interactive (CI) (Zhang et al., 2024), FSS-1000 (F) (Li et al., 2020), SA-1B (SA) (Kirillov et al., 2023), PartImageNet (PIN) (He et al., 2022), HumanParsing (HP) (Liang et al., 2015b;a), GoldG (GG) (Kamath et al., 2021).

| Method | Text Prompt Encoder | SAM? | Training Data | RefCOCO | | | RefCOCO+ | | | RefCOCOg | | AVG |
|---|---|---|---|---|---|---|---|---|---|---|---|---|
| | | | | val | testA | testB | val | testA | testB | val | test | |
| LAVT (Yang et al., 2022) | BERT-B | ✗ | RC, gRC | 72.7 | 75.8 | 68.8 | 62.1 | 68.4 | 55.1 | - | - | - |
| PolyFormer-L (Liu et al., 2023c) | BERT-B | ✗ | RC, gRC | 76.9 | 78.5 | 74.8 | 72.2 | 75.7 | 66.7 | 71.2 | 71.2 | 73.4 |
| UNINEXT-H (Yan et al., 2023) | BERT-B | ✗ | O, C, RC, V | 82.2 | 83.4 | 81.3 | 72.5 | 76.4 | 66.2 | 74.4 | 76.4 | 76.6 |
| UniLSeg-100 (Liu et al., 2023e) | CLIP-B | ✗ | SA, RC, gRC | 81.7 | 83.2 | 79.9 | 73.2 | 78.3 | 68.2 | - | - | - |
| UniRef++-L (Wu et al., 2023) | BERT-B | ✗ | RC, F, V | 79.1 | 82.1 | 77.5 | 68.4 | 74.0 | 61.5 | 71.4 | 72.8 | 73.4 |
| LISA (Lai et al., 2023) | Vicuna-7B | ✓ | A, CS, RC, PL, PP | 74.1 | 76.5 | 71.1 | 62.4 | 67.4 | 56.5 | 66.4 | 68.5 | 67.9 |
| PixelLM (Ren et al., 2023) | LLaMA2-13B | ✗ | A, CS, RC, PL, M | 73.0 | 76.5 | 68.2 | 66.3 | 71.7 | 58.3 | 69.3 | 70.5 | 69.2 |
| PixelLLM (Xu et al., 2024) | T5-XL | ✓ | RC, GG | 76.9 | 78.5 | 74.4 | 69.2 | 72.1 | 64.5 | 70.7 | 72.4 | 72.3 |
| GLaMM (Rasheed et al., 2023) | Vicuna-7B | ✓ | G, RC | 79.5 | 83.2 | 76.9 | 72.6 | 78.7 | 64.6 | 74.2 | 74.9 | 75.6 |
| u-LLaVA (Xu et al., 2023) | Vicuna-7B | ✓ | A, CS, RC, PL, PV | 80.4 | 82.7 | 77.8 | 72.2 | 76.6 | 66.8 | 74.8 | 75.6 | 75.9 |
| PSALM (Zhang et al., 2024) | Phi-1.5 | ✗ | C, RC, CI | **83.6** | **84.7** | **81.6** | 72.9 | 75.5 | 70.1 | 73.8 | 74.4 | 77.1 |
| EVF-SAM | BEIT-3 | ✓ | RC | 82.1 | 83.7 | 80.0 | 75.2 | 78.3 | 70.1 | 76.8 | 77.4 | 78.0 |
| EVF-SAM | BEIT-3 | ✓ | RC, O, A, PP, PIN, HP | 82.4 | 84.2 | 80.2 | **76.5** | **80.0** | **71.9** | **78.2** | **78.3** | **79.0** |

tated. Except for RefCOCO+, all datasets include geometric expression (*e.g.*, '*on the left*'). Among different splits of testing datasets, 'testA' is human-centric, while 'testB' aims for common objects.

**Extra training datasets.** To further enhance the versatility of EVF-SAM, we employ multi-task unified training by expanding the training datasets by introducing Objects365 (Shao et al., 2019), ADE20K (Zhou et al., 2017b), PASCAL-Part (Chen et al., 2014), PartImageNet (He et al., 2022), and HumanParsing (Liang et al., 2015b). Therefore, EVF-SAM can handle various granularity of text-prompted segmentation, *e.g.*, semantic-level, instance-level, and part-level segmentation. We refer the readers to the appendix for more details about training with the extra multi-task datasets.

**Metrics.** The gIoU and the cIoU are the most commonly calculated metrics on referring expression segmentation benchmarks. The gIoU is the average intersection-over-unions (IoU) among all images in the test datasets, while the cIoU is the cumulative intersection over the cumulative union. If not specifically declared, we follow previous works and report the cIoU as the main metric.

### 4.2 IMPLEMENTATION DETAILS

Unless specified, we initialize the proposed EVF-SAM with the public weights of SAM-ViT-Huge [1] (Kirillov et al., 2023) and BEIT-3-Large [2] (Wang et al., 2022a). All models are trained on 4 NVIDIA L40s GPUs with mixed precision. We adopt DeepSpeed (Song et al., 2023) with ZeRO-2 for model parallel to optimize memory consumption. During training, the batch size of each GPU is 16 and we use gradient accumulation for 2 steps, therefore the total batch size per iteration is 128. We adopt AdamW (Loshchilov & Hutter, 2017) optimizer and set the initial learning rate to 1e-4 with a linear-decay schedule. We train all models for 15k iterations (nearly 1 day) and use the binary cross-entropy loss (BCE) and dice loss (the weight of both losses is 1.0).

### 4.3 MAIN RESULTS

We mainly report the cIoU metric of RefCOCO-series benchmarks and compare our proposed EVF-SAM with recent state-of-the-art methods in Tab. 2 The upper part of Tab. 2 presents traditional

---

[1]SAM: https://github.com/facebookresearch/segment-anything
[2]BEIT-3: https://github.com/microsoft/unilm

Table 3: **Ablation on fusion methods.** We evaluate the performance of using different pre-trained Multimodal Encoders in EVF-SAM, *e.g.*, CLIP from OpenAI (Radford et al., 2021) or Open-CLIP (Ilharco et al., 2021). $L_i$ denotes the $i$-th layer in the BEIT-3 model (totally 24 layers for BEIT-3-Large). Half of the layers are activated to assess the impact of the modality fusion stage on model performance. †: pre-trained models provided by OpenAI. ‡: pre-trained models provided by OpenCLIP.

| Encoder | Params | Text | Image | Modality Fusion | RefCOCO | | | RefCOCO+ | | | RefCOCOg | | AVG |
|---|---|---|---|---|---|---|---|---|---|---|---|---|---|
| | | | | | val | testA | testB | val | testA | testB | val | test | |
| *CLIP variants.* | | | | | | | | | | | | | |
| CLIP-Large† | 123M | ✓ | | - | 61.0 | 63.4 | 59.9 | 43.1 | 45.9 | 40.6 | 48.9 | 49.6 | 51.6 |
| CLIP-Large† | 428M | ✓ | ✓ | Late (Concat) | 67.4 | 68.9 | 64.4 | 50.5 | 54.6 | 46.7 | 55.1 | 56.2 | 58.0 |
| CLIP-Large‡ | 123M | ✓ | | - | 60.8 | 63.2 | 59.0 | 42.9 | 46.4 | 39.2 | 49.2 | 50.5 | 51.4 |
| CLIP-Large‡ | 428M | ✓ | ✓ | Late (Concat) | 66.1 | 67.8 | 63.1 | 49.8 | 51.9 | 44.1 | 54.1 | 55.0 | 56.5 |
| CLIP-Huge‡ | 302M | ✓ | | - | 61.7 | 64.2 | 60.1 | 44.2 | 47.8 | 40.2 | 49.6 | 50.9 | 52.3 |
| CLIP-Huge‡ | 986M | ✓ | ✓ | Late (Concat) | 66.3 | 68.2 | 64.3 | 49.8 | 53.5 | 45.1 | 55.4 | 56.7 | 57.4 |
| *Early-fused vision-language models.* | | | | | | | | | | | | | |
| ViLT | 133M | ✓ | | - | 61.0 | 63.0 | 60.0 | 42.5 | 45.4 | 39.5 | 49.3 | 49.5 | 51.3 |
| ViLT | 136M | ✓ | ✓ | Late (Concat) | 61.4 | 64.0 | 59.6 | 42.8 | 46.4 | 40.1 | 49.5 | 50.0 | 51.7 |
| ViLT | 136M | ✓ | ✓ | Early | 73.9 | 75.3 | 70.9 | 61.1 | 64.4 | 55.2 | 65.1 | 66.8 | 66.6 |
| BEIT-3-Large | 370M | ✓ | | - | 61.6 | 65.1 | 59.4 | 44.0 | 47.6 | 40.6 | 49.5 | 50.8 | 52.3 |
| BEIT-3-Large | 673M | ✓ | ✓ | Late (Concat) | 67.7 | 70.2 | 65.4 | 51.1 | 55.0 | 46.9 | 57.2 | 57.0 | 58.8 |
| BEIT-3-Large | 673M | ✓ | ✓ | Early ($L_1 \sim L_{12}$) | 80.6 | 82.2 | 78.8 | 72.4 | 75.7 | 66.7 | 73.7 | 75.0 | 75.6 |
| BEIT-3-Large | 673M | ✓ | ✓ | Early ($L_1 \sim L_{24}$) | 82.1 | 83.7 | 80.0 | 75.2 | 78.3 | 70.1 | 76.8 | 77.4 | 78.0 |

methods based on text encoders. Despite their advantages in terms of fewer parameters and faster inference speeds, these methods either achieve less competitive results or require vast amounts of data due to their lack of integration with foundation models. The methods listed in the lower portion of Tab. 2 are based on Large Multimodal Models (LMMs), achieving state-of-the-art (SOTA) performance but require significant computational resources. Our EVF-SAM achieves the highest average cIoU score across all RES benchmarks, using only limited data and manageable computation costs. Specifically, our EVF-SAM achieves SOTA performance on RefCOCOg (Nagaraja et al., 2016; Mao et al., 2016), predicating a stronger capability for handling longer text prompts than previous LMM-based models, which is counter-intuitive while showing the great potential of vision-language models for understanding instructions. In addition, the early fusion between the input image and text prompts can generate more informative embeddings than independent encoders as discussed in Sec. 3.1.

## 4.4 ABLATION STUDY

In this section, we conduct experiments to investigate the vision-language models for text-prompted SAM and study the effects of the designs of the proposed EVF-SAM. Unless specified, we mainly report the cIoU on testA of RefCOCO.

**Multimodal Encoder and fusion methods.** In Tab. 3, we explore the effects of different Multimodal Encoders, *e.g.*, CLIP, ViLT (Kim et al., 2021), and BEIT-3, and fusion methods, *e.g.*, late fusion or early fusion. As shown in Tab. 3, using a text-only encoder in EVF-SAM obtains limited segmentation performance on RefCOCO. Using Multimodal Encoders with both image and text inputs remarkably improves 4.5 cIoU, 4.6 cIoU, 4.0 cIoU, 1.0 cIoU, and 4.5 cIoU for CLIP-Large† (OpenAI [3]), CLIP-Large‡ (OpenCLIP [4]), CLIP-Huge‡ (OpenCLIP), ViLT, and BEIT-3, respectively. It demonstrates the superiority of using multimodal prompts (text and input image) and showcases that the image embeddings will also provide useful guidance for SAM to segment objects accurately. We further evaluate the effects of early fusion on ViLT and BEIT-3, which adopts modality fusions in all self-attention layers. Specifically, we adopt two settings for BEIT-3 to analyze, *e.g.*, fusions among former 12 layers ($L_1 \sim L_{12}$), and fusions among all layers ($L_1 \sim L_{24}$). Tab. 3 indicates that BEIT-3 with early fusion (fusing former 12 layers or fusing all 24 layers) significantly improves compared to late fusion or using text only. In addition, ViLT with early fusion also achieves 11.1 cIoU improvements compared to the baseline with text-only prompts, showing the effectiveness of

---

[3]OpenAI: https://github.com/openai/CLIP

[4]OpenCLIP: https://github.com/mlfoundations/open_clip

Table 4: **Ablations on trainable modules.** We mainly evaluate the effects of fine-tuning or freezing the Multimodal Encoder, the prompt encoder and mask decoder of SAM. '✓' denotes trainable, while '∗' denotes frozen.

| Multimodal Enc. | Prompt Enc. | Mask Dec. | cIoU |
|:---:|:---:|:---:|:---:|
| ∗ | ∗ | ✓ | 21.2 |
| ✓ | ∗ | ∗ | 82.9 |
| ✓ | ∗ | ✓ | 83.3 |
| ✓ | ✓ | ✓ | **83.7** |

Table 5: **Ablations on multimodal feature representation.** BEIT-3 contains two `[CLS]` tokens for visual and textual modalities. We also explore the effects of `AvgPool` and late fusion between two modalities.

| $[CLS]_{Text}$ | $[CLS]_{Image}$ | $AvgPool_{Image}$ | Fusion | cIoU |
|:---:|:---:|:---:|:---:|:---:|
| ✓ | | | - | 83.5 |
| | ✓ | | - | **83.7** |
| | | ✓ | - | 83.5 |
| ✓ | ✓ | | Concat | 83.2 |

Table 6: **Comparison of effects of different foundation models.** AVG represents the average metric across the 8 RefCOCO-series benchmarks.

| Multimodal Encoder | SAM | Params | RefCOCO | | | RefCOCO+ | | | RefCOCOg | | AVG |
|---|---|---|---|---|---|---|---|---|---|---|---|
| | | | val | testA | testB | val | testA | testB | val | test | |
| CLIP-Large | SAM-ViT-H | 1.08B | 61.0 | 63.4 | 59.9 | 43.1 | 45.9 | 40.6 | 48.9 | 49.6 | 51.6 |
| ViLT | SAM-ViT-H | 783M | 73.9 | 75.3 | 70.9 | 61.1 | 64.4 | 55.2 | 65.1 | 66.8 | 66.6 |
| BEIT-3-Base | SAM-ViT-H | 863M | 78.9 | 80.6 | 75.3 | 69.8 | 74.2 | 63.0 | 71.6 | 72.9 | 73.3 |
| BEIT-3-Large | Efficient-SAM-S | 700M | 82.5 | 83.5 | **80.4** | 75.4 | 77.9 | 70.2 | 76.1 | 77.1 | 77.9 |
| BEIT-3-Large | SAM-ViT-H | 1.32B | 82.1 | 83.7 | 80.0 | 75.2 | 78.3 | 70.1 | 76.8 | 77.4 | 78.0 |
| BEIT-3-Large | SAM-2-L | 898M | **82.7** | **84.1** | 80.0 | **76.3** | **80.1** | **71.8** | **77.0** | **78.4** | **78.8** |

early fusion and multimodal inputs for prompting SAM. Therefore, Tab. 3 demonstrates that *(1) Multimodal Encoder with the input image and text and (2) early fusions between the image and text encoder* are much effective for text-prompted SAM.

**Ablations on trainable modules.** In Tab. 4, we evaluated the effects of fine-tuning (✓) or freezing (∗) modules in the proposed EVF-SAM, *i.e.*, the Multimodal Encoder, the prompt encoder, and the mask decoder. The image encoder of SAM is kept frozen during training. As Tab. 4 shows, fine-tuning the Multimodal Encoder is crucial and it adapts the Multimodal Encoder to encode text and image inputs to multimodal representation for referring image segmentation. Notably, EVF-SAM can achieve competitive results with all modules of SAM kept frozen, and it can be seamlessly regarded as a strong extension for the original SAM, which simultaneously supports text prompts, box prompts and point prompts. Tab. 4 Further fine-tuning the prompt encoder and mask decoder of SAM brings significant improvements.

**Multimodal feature representation.** In Tab. 5, we explore the effects of using different multimodal features representations as prompts for SAM. Specifically, we adopt different outputs of the Multimodal Encoder: (a) the image `[CLS]` token, (b) the `AvgPool` over image tokens, and (c) the text `[CLS]` token. Tab. 5 shows that using image `[CLS]` token is more effective while combining image and text tokens through concatenation leads to a performance drop.

**Effects of Different Foundation Models.** In Tab. 6, we explore the effects of using different foundation models in EVF-SAM. For the Multimodal Encoder, we adopt CLIP-Large (only text encoder), ViLT, BEIT-3-Large, and BEIT-3-Base. We also modify EVF-SAM with Efficient-SAM (Xiong et al., 2023) to formulate a lighter version, which reduces 600M parameters compared to SAM-H. As shown in Tab. 6, EVF-SAM with BEIT-3-Base brings a severe performance drop which indicates a better Multimodal Encoder leads to better prompts for SAM. Remarkably, Tab. 6 shows a negligible difference between Efficient-SAM-S and SAM-H in EVF-SAM, which demonstrates the effectiveness of Efficient-SAM and also indicates that EVF-SAM performs well for different SAM variants. In addition, it also provides insights about designing text-prompted SAMs for future research, *e.g.*, *developing a larger and better Multimodal Encoder is more important to empower SAM with text-prompted abilities*.

## 4.5 DISCUSSIONS

To unveil how the multimodal encoder contributes to prompting SAM with texts, we visualize the attention maps between the `[CLS]` token (prompt embeddings) and the image tokens from the last layer of BEIT-3. As shown in Fig. 5, the attention maps focus on the target objects and are consistent with the input text prompts. The deep fusion of text and image embeddings leads to accurate region-

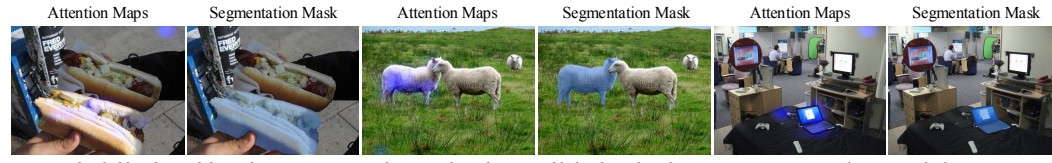

| Attention Maps | Segmentation Mask | Attention Maps | Segmentation Mask | Attention Maps | Segmentation Mask |

*" loaded hot dog with bite taken out "*     *" sheep standing close to and behind another sheep "*     *" a laptop on a bed "*

Figure 5: **Visualizations of Attention Maps in Multimodal Encoder.** To unveil the effects of the Multimodal Encoder, we visualize the attention maps between the [CLS] token and image tokens in the last layer of BEIT-3-Large. Specifically, we sum up the attention maps from all heads.

text alignment. Consequently, the prompt embeddings contain abundant object-related information, including semantics and spatial localization, which is conducive to SAM achieving precise object segmentation.

## 5 CONCLUSION

In this paper, we have explored the effective ways to prompt SAM with texts and demonstrate the importance of using the Multimodal Encoder with early fusion and multimodal inputs, *i.e.*, text prompts and input images. To this end, we propose EVF-SAM, which establishes a new and simple path for extending SAMs' text-prompted segmentation abilities with the *off-the-shelf* foundation models. We conduct experiments on the referring expression segmentation (RES) tasks with various benchmarks to evaluate the performance of text-prompted SAM. Experimental results showcase that our EVF-SAM achieves state-of-the-art performance for segmenting objects with referring texts on RefCOCO/+/g benchmarks, outperforming recent approaches based on Large Language Models with huge numbers of parameters. Moreover, experiments prove that (1) a multimodal encoder with input text and image and (2) the early fusion between image and text do matter more for prompting SAM than vanilla text encoders or Large Language Models. We hope this study and experiments can bring new ideas or insights to inspire future research on prompting SAM with texts.

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

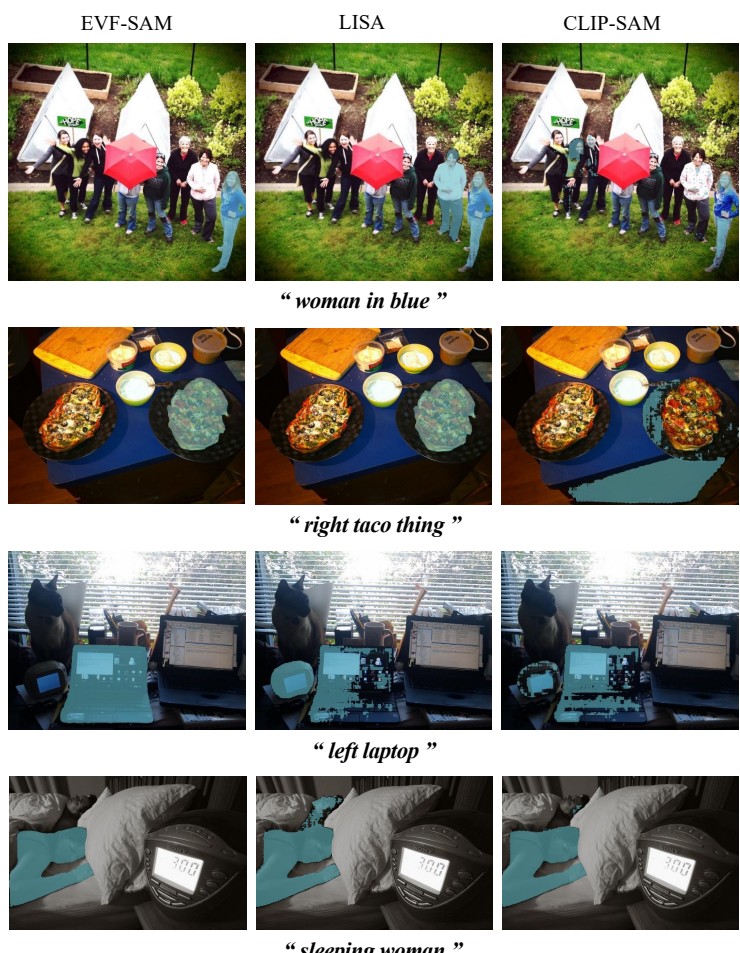

Figure 6: **Visualization Results on RefCOCO val.** We compared the qualitative results on Ref-COCO which contains simple descriptive expressions.

# A APPENDIX

## A.1 QUALITATIVE RESULTS

In this section, we mainly visualize the qualitative results on RefCOCO *val* and RefCOCOg *val* datasets, as shown in Fig. 6 and Fig. 7, respectively. Moreover, we compare the qualitative results of different ways to prompt SAM with texts: (1) our proposed EVF-SAM, (2) SAM with LLM (LISA (Lai et al., 2023)), and (3) SAM with a CLIP text encoder implemented in this paper (suggested by (Kirillov et al., 2023), which are based on the same SAM-Huge model. The qualitative results can demonstrate the superiority of the proposed EVF-SAM.

**Visualizations on RefCOCO.** Fig. 6 shows the qualitative comparisons on the RefCOCO *val*, which contains simple *descriptive* expression texts. The proposed EVF-SAM can follow the expressions and segment more accurately with clear boundaries.

**Visualizations on RefCOCOg.** Fig. 7 illustrates the qualitative comparisons on the RefCOCOg *val*, which aims to segment objects with *long* expression texts. The SAM with a vanilla CLIP text encoder produces inferior segmentation results given the long-expression texts. However, the proposed EVF-SAM outperforms LISA when using long expressions, even though LISA adopts LLaMA-7B (Touvron et al., 2023) to understand the instructions and generate prompt embeddings, showcasing that the lightweight vision-language models can understand complex expressions. In

addition, the proposed EVF-SAM can also understand the texts or expressions towards spatial locations, such as *'the umbrella closest to the camera'*.

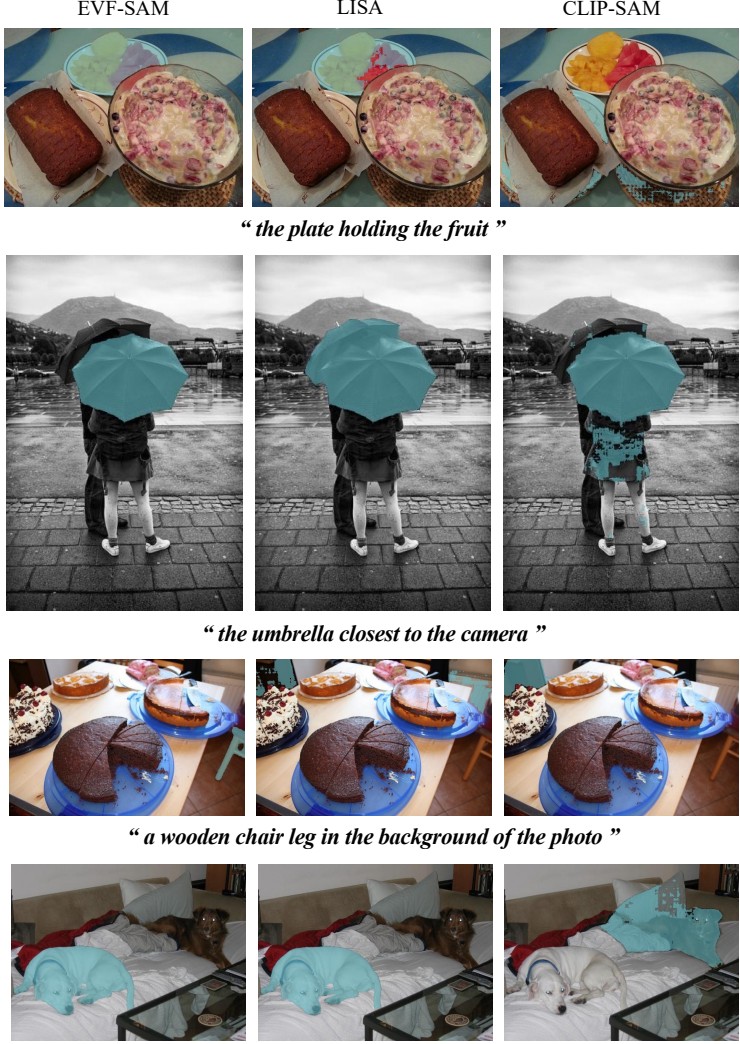

Figure 7: **Visualization Results on RefCOCOg val.** Considering that RefCOCOg contains longer expressions and we provide qualitative results to show the capability of our EVF-SAM for understanding long expressions.

## A.2 TRAINING EVF-SAM WITH MULTI-TASKS

To further enhance the generic capability of our EVF-SAM, we propose to implement multi-task training. Based on the experiments that show the performance degradation when simply including extra segmentation data, we explore ways to make our EVF-SAM gain from extra data.

**Mixed training with semantic segmentation.** We introduce some extra semantic segmentation datasets (ADE20K (Zhou et al., 2017b), Mapillary (Neuhold et al., 2017)) to proceed with joint training. We do not include COCO-Stuff (Caesar et al., 2018) to avoid data leakage with Ref-COCO/+/g. It can be seen in Tab. 7 that the performance on RefCOCO+ and ADE20K gains, indicating the effectiveness of including extra data to enhance the generic capability. However, the evaluation metrics of RefCOCO and RefCOCOg decrease when simply including extra semantic segmentation data. We owe this phenomenon to semantic conflict (Liu et al., 2023e).

Table 7: **Results of adding extra semantic data.** * means zero-shot results. The reported ADE20K results are evaluated on the validation set using the cIoU metric.

| ADE20K | Mapillary | RefCOCO | | | RefCOCO+ | | | RefCOCOg | | ADE20K |
|---|---|---|---|---|---|---|---|---|---|---|
| | | val | testA | testB | val | testA | testB | val | test | val |
| | | **82.1** | **83.7** | 80.0 | 75.2 | 78.3 | 70.1 | **76.8** | 77.4 | 54.2* |
| ✓ | | 81.7 | 83.6 | **80.3** | 75.4 | **78.4** | **71.3** | 75.5 | **77.6** | 75.9 |
| | ✓ | 81.9 | 83.5 | **80.3** | 75.1 | 78.0 | 70.8 | 75.3 | 77.4 | 59.6* |
| ✓ | ✓ | 81.8 | 83.4 | 79.7 | **75.6** | 78.0 | 70.7 | 75.8 | 76.9 | **76.1** |

**Unified training with multi-task datasets.** To solve the semantic conflict mentioned above, we propose several pre-process strategies for datasets of different distributions. We will open-source related codes in our project page.

- *Instance-level data:* We apply Objects365 (Shao et al., 2019) to extend RES data. Specifically, (a) for each image, we exclude categories with more than one instance to avoid ambiguity problem. (b) we employ SAM-2 (Ravi et al., 2024) to automatically annotate masks according to the selected ground-truth bounding boxes. The remaining annotations maintain a rich amount thanks to the dense annotation of Objects365 (Shao et al., 2019). We obtain 524K images (of original 600K images) with 1.8M annotations (of original 10M annotations). The mask quality from automatic annotation is fine thanks to the accurate ground-truth from Objects365 (Shao et al., 2019) and the powerful segmentation capability of SAM-2 Ravi et al. (2024). Besides, the remaining annotations are valuable for addressing long-tail problems because those excluded annotations often belong to head categories.

- *Semantic-level data:* We introduce ADE20K (Zhou et al., 2017a; 2019) to broaden multi-task capability. We construct a special token '[semantic]' and input '*[semantic] {category}*'. The special token would not be limited to common grammar so it is helpful to avoid semantic conflict.

- *Part-level data:* To enable the model to segment parts of objects, we introduce PartImageNet (He et al., 2022), HumanParsing (Liang et al., 2015a;b) and PASCAL-Part (Chen et al., 2014) to train our model. For semantic-level annotated datasets, *i.e.*, HumanParsing, we implement the same strategy as ADE20K. Exceptionally, we align the definition of 'left' and 'right' with RES datasets (*e.g.*, RefCOCO). For instance-level annotated datasets, *i.e.*, PartImageNet and PASCAL-Part, we merge instance masks of the same category to convert the dataset to semantic-level. Then, the same strategy as ADE20K is implemented.

By combining those datasets, we observe a significant performance gain of 1.0 cIoU on the average metric, as shown in Tab. 2. Moreover, our model is able to proceed with multiple tasks like part-segmentation and semantic-level segmentation.

