# OpenReview forum: "EVF-SAM: Early Vision-Language Fusion For Text-Prompted Segment Anything Model"
_ICLR.cc/2025/Conference — ICLR 2025 Conference Withdrawn Submission_

### Official Review · Reviewer_3Ljb · 2024-10-16

**Soundness:** 2
**Presentation:** 3
**Contribution:** 2
**Rating:** 5
**Confidence:** 3

**Summary:**

This paper introduces EVF-SAM, which enhances the text-prompted segmentation ability of SAM by (1) leveraging a stronger multimodal encoder and (2) implementing early vision-language fusion. Through these techniques, it achieves new state-of-the-art performance on referring image segmentation tasks.

**Strengths:**

1. The proposed method achieves state-of-the-art performance in referring segmentation.
2. The paper is well-written and easy to read.

**Weaknesses:**

1. Novelty:
> Using a stronger backbone (BEIT vs. CLIP) and early fusion are both expected to yield better performance, and it's unclear what additional insights this paper provides. Early or late fusions of multimodal inputs have been explored in many earlier referring segmentation works, such as LAVT[1], CoupAlign[2], and MaskGrounding[3], which all reach similar conclusions as this paper regarding the superiority of early fusion. The authors should clarify the unique insights gained from this paper.

2. Clarity:
> 1. Given that early fusion is presented as a core contribution, a detailed explanation of the process is expected. However, this information seems to be missing from the paper. The only details provided are a high-level diagram in Fig. 3 and unclear wording in line 230, which states, "We leverage the 'early-fusion' vision-language model as the Multimodal Encoder to generate prompt embeddings for SAM." It's unclear whether early fusion is a proposed novelty or part of the original BEIT-3 architecture. It appears to be the latter, suggesting that the main change is swapping CLIP with BEIT-3. Is that true?
> 2. In lines 237-240, the authors claim that prompt embeddings with special tokens lead to "instability during both training and inference." However, it seems that EVF-SAM also uses a special token (the [CLS] token after projections) as prompt embeddings, and it's unclear how this differs and resolves the previously mentioned problems.
> 3. For Tab. 7, providing mIoU results for ADE20K would be beneficial, as cIoU is not commonly used for this dataset, making comparisons with existing methods difficult.

3. Unfair comparisons:
> 1. The authors claim that EVF-SAM is superior to decoder-only Large Language Models like LAVT and PSALM based solely on referring segmentation performance. However, these LLM-based models are capable of more than just referring segmentation, including reasoning segmentation, generic segmentation, VQA, etc. This comparison is unfair unless the authors can demonstrate superiority in these other tasks as well.

4. Insufficient ablations:
> 1. Based on Tab. 3, early fusion experiments using the widely adopted CLIP model are lacking, and the improvements seen with early fusion in ViLT are modest (average 66.6 vs. 51.7). In contrast, for BEIT-3-Large, the improvement is significant, with performance increasing from an average of 52.3 to 78.0. This suggests that the early fusion mechanism may be particularly effective for BEIT-3-Large but may not be universally beneficial across other vision-language models. Given this, can the strong performance in Tab. 2 still be achieved with the more commonly used CLIP model?
> 2. The experiments in Tab. 2 are done using different training data, which makes fair comparison difficult. It will be great if the author can provide fairer comparisons using same training data as other works such as PSALM and LISA.

References:

[1] LAVT: Language-Aware Vision Transformer for Referring Image Segmentation, CVPR 2022

[2] CoupAlign: Coupling Word-Pixel with Sentence-Mask Alignments for Referring Image Segmentation, NeurIPS 2022

[3] Mask Grounding for Referring Image Segmentation, CVPR 2024

**Questions:**

1. Is the early fusion mechanism also applicable to other types of architectures, such as the decoder-only LLM-based architectures like PSALM and LISA?
2. It will be greatly beneficial to the community if the author can provide code and data to reproduce the experiments in this paper.

---

### Official Review · Reviewer_neof · 2024-10-27

**Soundness:** 3
**Presentation:** 3
**Contribution:** 2
**Rating:** 5
**Confidence:** 4

**Summary:**

This paper continues the research direction on text-prompted segmentation foundation models and proposes EVF-SAM with an early fusion paradigm. The authors find that multimodal prompts and early-fusion VLMs are beneficial for referring expression segmentation. Extensive experiments show the superiority of the proposed EVF-SAM model with much fewer parameters.

**Strengths:**

1. The findings of early-fusion multimodal encoder models are intriguing, which is helpful for the community when designing text-prompted segmentation models.

2. The experiments are comprehensive, with detailed empirical evaluations of the design choices of each part in the proposed method. The improvements are significant.

**Weaknesses:**

1. The baseline of LLaVA is not fair enough since LISA mainly focuses on its Reasoning Segmentation ability. So it is necessary to fine-tune LLM-based models like LISA, GSVA with referring expression segmentation tasks to achieve a similar level.

2. The training samples of LISA-7B models are 2 img/gpu $\times$ 8 gpu $\times$ 10 grad-accumulation $\times$ 500 steps/epoch $\times$ 10 epoch = 800k, however EVF-SAM has 128 samples/iter $\times$ 15k steps = 1,920k samples. The number of training samples should also be ablated carefully.

3. The novelty is somehow limited since the architecture and training methods of incorporating VLMs into RES, REC, GRES tasks have been investigated by previous works such as LISA, GSVA, PixelLM, GLaMM, u-LLaVA, PSALM. Using another early-fusion multimodal encoder seems not have enough innovation in terms of methodology. However, considering the abundant experiment results, it is not appropriate to deny the contribution of this paper. The authors are encouraged to provide more theoretical analyses and insights to justify why the early-fusion paradigm encoders bring such significant improvements.

**Questions:**

Please refer to the weaknesses section.

---

### Official Review · Reviewer_qfP9 · 2024-10-30

**Soundness:** 2
**Presentation:** 1
**Contribution:** 2
**Rating:** 3
**Confidence:** 4

**Summary:**

Authors provide two main contributions in effectively extending SAM's general segmentation capability to Referring Expression Segmentation (RES). Firstly, they empirically show that prompting SAM with a multimodal prompts and employing 'early fusion' of image and text embeddings to produce such prompts is better than using unimodal, text prompts and employing 'late fusion'. Secondly, they propose EVF-SAM, which employs BEIT-3 as a multimodal encoder that produces a multimodal prompt from a downscaled version of the input image and referring text and inputs the multimodal prompt into SAM's original prompt encoder. Training the multimodal encoder and finetuning SAM's prompt encoder and mask decoder, EVF-SAM acquires strong performance on major RES benchmarks, especially on the longer and more difficult RefCOCO+ and RefCOCOg datasets given additional segmentation datasets for training. Authors highlight that their EVF-SAM is not only better in performance but also computationally efficient compared to LLM-based models for the smaller architecture of the multimodal encoder compared to LLMs.

**Strengths:**

- Authors show that using a comparatively lightweight multimodal encoder can provide stronger performance than using heavier LLMs.
- Authors show that investing in building an effective multimodal encoder may be more effective than building a more effective segmentation model (larger BEIT-3 is better than using larger SAM illustrated in "4.4 Ablations: Effects of Different Foundation Models.").

**Weaknesses:**

- Lack of novelty
  - The proposed framework seems to be an addition of BEIT-3 as an multimodal encoder to SAM. No new modules nor techniques, but simply adding an existing, well-known multimodal encoder to another existing, well-known segmentation model.

- Omitted references
   - When explaining previous work that explore the text-prompted abilities of SAM (line 051~), authors explain each work but omits references to them. Thus, the reviewer cannot be sure of these findings they draw from these work are valid.
   - Findings used to justify the method also lacks references: ie. these sentences do not have references "However, the uncontrolled
length of the answering query introduces instability during both training and inference. Forcing the model to conform to a specific answering template can lead to language drift."

- Unclear writing
  - Authors should better explain what they mean by "matches" in "Fast-SAM (Zhao et al., 2023) matches the similarity of CLIP (Radford et al., 2021) features between the text and Region of Interest (RoI) of image."
  - Authors simply list these short sentences without sufficient explanation in Section 2.2: "PerceptionGPT (Pi et al., 2023) proposes an end-to-end architecture. u-LLaVA (Xu et al., 2023) supports multi-task. PSALM (Zhang et al., 2024) imports mask tokens to LMM input for better performance."
  - Authors claim that traditional RES models are limited since they "They fail to integrate with large-scale foundation models... e.g., SAM, LLaVA," but methods that employ LLMs and LMMs have "heavy computation burden", which seem contradictory. I understand that authors wanted to explain that traditional RES models are not using foundation segmentation/vision models and thus missing out on the generalizablity of these models and that models that use LLMs are inherently computationally heavy, but using LLaVA (which is LLM-based) as an example for the first point as a model that traditional models *should* try to incorporate and then claiming LLM-based models are heavy is inconsistent.
  - Authors explain that they use a special [semantic] token to alleviate semantic conflict but do not explain what the conflict is. Although I acknowledge the reference to UniLSeg, I think a short explanation would be very helpful.

**Questions:**

- Why do the authors use "zero-initialized sparse embeddings" to concatenate to multimodal embeddings before feeding into SAM's prompt encoder. Is there a specific reason?
- Why does the prompt encoder in SAM need to be trained or even included in this framework? Can we not simply input the multimodal embedding from the BEIT-3 directly into the mask decoder? What is the role of SAM's prompt encoder? Given the minor performance increase with finetuning SAM's prompts encoder (0.4 cIoU in Table 4) compared to freezing it, what is intention in finetuning it?
- Why is using the image [CLS] token most effective in Table 5?
- In Figure 5, the [CLS] token seems to attend to right upper image tokens (which are unrelated to the referring text nor salient parts of the image) in both the first and third attention maps. This may be explanable by this paper "Vision Transformers need Registers, Darcet et al., ICLR 2024" so checking it out may help.
- Is there a possibility that BEIT-3 (Image+Text) does better than CLIP (Image+Text) because it has more computation in Table 1? If BEIT-3 (Image+Text) has less or similar computation, the conclusion that early fusion is better than late fusion would be more convincing.

---

### Official Review · Reviewer_s4fQ · 2024-11-01

**Soundness:** 2
**Presentation:** 3
**Contribution:** 2
**Rating:** 3
**Confidence:** 4

**Summary:**

This paper proposes EVF-SAM, a novel approach to enhance SAM’s text-based segmentation capabilities by leveraging multimodal prompts and early vision-language fusion. EVF-SAM integrates a multimodal encoder (e.g., BEIT-3) with SAM through a simple projector, transforming text and image inputs into a unified embedding optimized for segmentation tasks. Using instruction-free templates and a unified multi-task training strategy, EVF-SAM significantly outperforms prior SAM methods on RES datasets (e.g., RefCOCO/+/g) with only 1.32 billion parameters, achieving state-of-the-art results while reducing model size by nearly 82%. This method demonstrates the advantages of early multimodal fusion for referring expression segmentation.

**Strengths:**

1.	The paper presents a approach to enhancing SAM’s text-prompted segmentation through EVF-SAM, which leverages multimodal encoding and early vision-language fusion. The originality lies in tackling the relatively unexplored area of text-guided segmentation in SAM, addressing significant limitations in existing grounded detectors and large language models for this task. By utilizing an encoder-based approach with early fusion, the method efficiently merges text and image modalities, ensuring a more cohesive representation and improving SAM’s capability for referring expression segmentation without the computational burden of LLMs or the need for manual prompt design.
2.	In terms of quality, the proposed method is rigorously evaluated on multiple RES datasets (RefCOCO, RefCOCO+, RefCOCOg), where it demonstrates state-of-the-art performance with a highly efficient architecture, achieving notable gains with a reduced parameter count of 1.32B, approximately 82% fewer parameters than comparable methods.
3.	Finally, the significance of this work is good. By demonstrating a feasible, efficient way to incorporate text prompts in SAM through multimodal encoding and early fusion, the paper not only advances SAM’s application in multimodal tasks but also contributes valuable insights to the broader field of vision-language fusion.

**Weaknesses:**

1.	The authors suggest that using a Multimodal Encoder with early fusion enhances SAM’s text-prompting capability. However, the study validates only two multimodal models—CLIP and BEIT-3. The efficacy of multimodal input in these cases could stem from the models’ unique training strategies, such as BEIT-3’s use of shared multi-head self-attention. It remains unproven whether other multimodal models, such as VLMO [1] and CoCa [2], would also benefit from this early fusion and Multimodal Encoder strategy. Further evaluation is needed to determine if these approaches can enhance SAM’s text capability across a broader range of multimodal models.
2.	The authors state that “Despite their advantages in terms of fewer parameters and faster inference speeds, these methods either achieve less competitive results or require vast amounts of data due to their lack of integration with foundation models.” However, in Table 2, there is no clear indication of the parameter counts or training data volumes for the comparison methods, making it difficult to assess the efficacy of the proposed approach. To strengthen the results, the authors should explicitly report parameter counts, training data volumes, and inference speeds for the comparative methods.
3.	The authors validate their method solely on the referring expression segmentation benchmark. With the introduction of text capability in SAM, there is a risk that EVF-SAM might overfit this specific benchmark, potentially reducing its effectiveness on unseen or less structured data. To demonstrate the robustness of EVF-SAM’s text-prompting ability, additional validation across multiple tasks, such as zero-shot semantic segmentation, zero-shot instance segmentation, and open-vocabulary segmentation, would be added.
4.	The paper primarily compares EVF-SAM against methods specific to the referring expression segmentation task. To provide a comprehensive comparison, the authors should also consider benchmarking against other methods aimed at enhancing SAM’s text capability, such as Open-Vocabulary SAM [3], SAM-CLIP [4], and AlignSAM [5].
5.	In the related work section, the authors should include a broader summary of research focused on SAM’s limitations and methods for enhancing its various capabilities. Relevant works include [6], [7], [8], and [9].

[1]Bao, Hangbo, et al. "Vlmo: Unified vision-language pre-training with mixture-of-modality-experts." Advances in Neural Information Processing Systems 35 (2022): 32897-32912.
[2]Yu, Jiahui, et al. "Coca: Contrastive captioners are image-text foundation models." arXiv preprint arXiv:2205.01917 (2022).
[3]Yuan, Haobo, et al. "Open-vocabulary sam: Segment and recognize twenty-thousand classes interactively." European Conference on Computer Vision. Springer, Cham, 2025.
[4]Wang, Haoxiang, et al. "Sam-clip: Merging vision foundation models towards semantic and spatial understanding." Proceedings of the IEEE/CVF Conference on Computer Vision and Pattern Recognition. 2024.
[5]Huang, Duojun, et al. "Alignsam: Aligning segment anything model to open context via reinforcement learning." Proceedings of the IEEE/CVF Conference on Computer Vision and Pattern Recognition. 2024.
[6]Xie, Defeng, et al. "Edit everything: A text-guided generative system for images editing." arXiv preprint arXiv:2304.14006 (2023).
[7]Sun, Yanpeng, et al. "VRP-SAM: SAM with visual reference prompt." Proceedings of the IEEE/CVF Conference on Computer Vision and Pattern Recognition. 2024.
[8]Yao, Jingfeng, et al. "Matte anything: Interactive natural image matting with segment anything model." Image and Vision Computing 147 (2024): 105067.
[9]Yu, Tao, et al. "Inpaint anything: Segment anything meets image inpainting." arXiv preprint arXiv:2304.06790 (2023).

**Questions:**

The questions raised in this section are the same as the weaknesses outlined above.

---

### Official Review · Reviewer_QY3Y · 2024-11-03

**Soundness:** 2
**Presentation:** 3
**Contribution:** 2
**Rating:** 5
**Confidence:** 4

**Summary:**

The paper proposes EVF-SAM for referring segmentation on text-prompt based segment anything model. EVF_SAM explores both multimodal prompts and vision-language models with early fusion to prompt SAM. The papers compared the early fusion to the late modality fusion manners. Experiments are conducted on RefCOCO, RefCOCO+ and RefCOCOg to show its higher performance and fewer model parameters.

**Strengths:**

1. The paper is written in a clear structure and organized well. The figures have a nice summarization over existing text-prompt manners (fig1) and the models's architecture (fig 4).

2. The paper has a good ablation experiment analysis. Table 3 provides a comprehensive ablation experiment on various fusion methods, which shows the advantage brought by early fusion for vision-language models and supports paper's main claim. Table 4, 5 and 6 provide answers for trainable modules, feature representation and foundation model design choice.

3. The paper shows a strong result on RefCOCO, RefCOCO+ and RefCOCOg as in Figure 1.

**Weaknesses:**

1. The paper has a limited tech novelty and contribution. Early cross-modal fusion between text and image features using cross-attention has been widely explored in popular language grounding methods, such as Grounding DINO, Grounding DINO1.5 [a] and GLIP [b]. There is no clear discussion on the differences to these existing methods. The language grounding tasks and referring segmentation tasks are very close related.

2. The paper only evaluates on the COCO related datasets, more in-the-wild testing on more diverse datasets will also be appreciated, such as using the ReasonSeg proposed by LISA or ReasonSeg-Ins by LISA++.

3. In Table 2, all comparing method are using different text-prompt based encoders, this makes the results comparison not that fair. As BEIT-3 is not the contribution of the paper. If using ViLT as the encoder, EVF-SAM's model performance will decrease from 82.1 to 73.9 on Ref-COCO val set. This means the SOTA performance of EVF-SAM is highly dependent on pretrained BEIT-3 while the comparing methods in Table 2 didnot use it.

4. Besides model parameter comparison, a comparison on the running inference speed in Table 6 will help readers understand better.

**Questions:**

More results comparison on different datasets and a detailed difference discussion to existing language grounding methods on early-fusion will make the paper's contribution more clear.

---

### Note · Authors · 2024-11-13

I have read and agree with the venue's withdrawal policy on behalf of myself and my co-authors.